# Predictors of recurrence of major depressive disorder

**Munn-Sann Lye** [1] *, **Yin-Yee Tey**[1], **Yin-Sim Tor**[1¤], **Aisya Farhana Shahabudin**[1], **Normala Ibrahim** [2], **King-Hwa Ling**[3], **Johnson Stanslas**[4], **Su-Peng Loh**[5], **Rozita Rosli**[3], **Khairul Aiman Lokman**[5], **Ibrahim Mohammed Badamasi** [4], **Asraa Faris-Aldoghachi**[3], **Nurul Asyikin Abdul Razak**[2]

**1** Department of Community Health, Faculty of Medicine and Health Sciences, Universiti Putra Malaysia, Selangor, Malaysia, **2** Department of Psychiatry, Faculty of Medicine and Health Sciences, Universiti Putra Malaysia, Selangor, Malaysia, **3** Department of Biomedical Science, Faculty of Medicine and Health Sciences, Universiti Putra Malaysia, Selangor, Malaysia, **4** Pharmacotherapeutics Unit, Department of Medicine, Faculty of Medicine and Health Sciences, Universiti Putra Malaysia, Selangor, Malaysia, **5** Department of Nutrition and Dietetics, Faculty of Medicine and Health Sciences, Universiti Putra Malaysia, Selangor, Malaysia

¤ Current address: School of Bioscience, Faculty of Health and Medical Sciences, Taylor's University, Subang Jaya, Malaysia
* lyems9@yahoo.com

**Data Availability Statement:** All relevant data are within the manuscript and its Supporting Information files.

**Funding:** The study was funded by Research Management Centre Universiti Putra Malaysia (GP-

## Abstract

A total of 201 patients with major depressive disorder from four hospitals in Malaysia were followed up for 5 years to determine the prognostic factors of recurrent major depressive disorder that could potentially contribute to improving the management of MDD patients. For each individual patient, at the time of recruitment as part of a case-control study, information was collected on recent threatening life events, personality and social and occupational functioning, while blood samples were collected to genotype single nucleotide polymorphisms of vitamin D receptor (VDR), zinc transporter-3 (ZnT3), dopamine transporter-1 (DAT1), brain-derived neurotropic factor (BDNF), serotonin receptor 1A (HT1A) and 2A (HT2A) genes. Kaplan-Meier and Cox-regression were used to estimate hazard functions for recurrence of major depressive disorder. Individuals with severe MDD in previous major depressive episodes had five and a half times higher hazard of developing recurrence compared to mild and moderate MDD (HR = 5.565, 95% CI = 1.631–18.994, p = 0.006). Individuals who scored higher on social avoidance had three and a half times higher hazard of recurrence of MDD (HR = 3.525, 95% CI = 1.349–9.209; p = 0.010). There was significant interaction between *Apa*I +64978C>A single nucleotide polymorphism and severity. The hazard ratio increased by 6.4 times from mild and moderate to severe MDD for A/A genotype while that for C/A genotype increased by 11.3 times. Social avoidance and severity of depression at first episode were prognostic of recurrence. Screening for personality factors at first encounter with MDD patients needs to be considered as part of the clinical practice. For those at risk of recurrence in relation to social avoidance, the psychological intervention prescribed should be customized to focus on this modifiable factor. Prompt and appropriate management of severe MDD is recommended to reduce risk of recurrence.

IPB/2013/9415700). The funders had no role in study design, data collection and analysis, decision to publish, or preparation of the manuscript.

**Competing interests:** The authors have declared that no competing interests exist.

## Introduction

Major depressive disorder (MDD) is defined as "discrete episodes of at least 2 weeks' duration with clear-cut changes in affect, cognition, and neuro-vegetative functions, and inter-episode remissions" and often is recurrent in nature [1]. Recurrence is the return of symptoms after at least 2 consecutive months between separate episodes during which time criteria are not met for a major depressive episode (MDE) and there must be the return of at least 5 out of 9 symptoms of depression [1]. Many patients are at substantial risk of later recurrence, with 60% lifetime risk of recurrence after the first major depressive episode. As many as 70% of those with 2 MDEs have recurrences throughout their life, and 90% of those with three or more episodes will experience further recurrent episodes [2]. In addition, one third to half of the patients had recurrence within one year of discontinuation of treatment [3]. Each recurrence also carries a 10–20% risk of becoming unremitting and chronic [4]. Recurrent MDD in turn increases risk of significant functional impairment, suicide and comorbid physical health problems [5–9], incurring heavy health and economic burdens [10].

Many studies showed that sociodemographic factors such as younger age, women, and those who have never been married were at higher risk of recurrence [7, 11–15]; however, results of other studies on these risk factors have been inconsistent. A systematic review by Hardeveld and colleagues (2009) did not find any association between sociodemographic factors such as age, gender, socioeconomic status, marital status and risk of recurrence [16, 17, 18]. However, clinical variables such as number of previous episodes [7, 11, 19–22], severity of last episode [21, 22] and family history of MDD [21– 23] were strong predictors of recurrence. In addition, patients who had residual depression symptoms despite responding to treatment, had higher risk of recurrence (Serafini *et al.*, 2018; Nierenberg *et al.*, 2010; Judd *et al.*, 1998) [19, 24, 25]. Studies also showed that personality traits were related to risk of recurrence [26]; Nobbelin *et al.* (2018) reported that patients with premorbid nervous or tense personalities had greater likelihood of recurrence [27] (Nobbelin *et al.*, 2018).

Burcasa and Iacono in their review suggested that vulnerability to recurrent MDD has a genetic component [28]. Mutations in the brain-derived neurotropic factor (BDNF) [29, 30] and serotonin receptor genes [31] were shown to be associated with recurrent MDD. In addition, findings by Kuningas *et al.* (2009) [32] and Lye *et al.* [33] that single nucleotide polymorphisms (SNPs) of vitamin D receptor (VDR) and zinc transporter-3 (SLC30A3) genes were respectively associated with MDD, and together with evidence that SNPs of dopamine transporter-1 gene increased risk of MDD[34, 35], led to renewed interest in determining if an association exists between VDR, ZnT3 SNPs and recurrence of MDD.

Although acute management of MDDs is generally effective, many patients experience recurrence following remission. Identifying prognostic factors in patients at high risk of recurrence provides a window of opportunity for specific secondary prevention as well as initiation of long-term maintenance treatment. The objective of the present study is to determine the prognostic factors of recurrence that could potentially contribute to improving the management of patients with MDD.

## Materials and methods

### Ethics statement

This study was approved by the Medical Research Ethics Committee of the Ministry of Health Malaysia (NMRR No.: NMRR-14-688-19696). Written informed consent was obtained from all the participants.

## Data collection

The original study was a case-control study in which the methodology has been described elsewhere [33]. Inclusion criteria included male and female patients from 18 to 65 years of age, who were diagnosed with single or recurrent, non-psychotic episode of Major Depressive Disorder (using the Diagnostic and Statistical Manual of Mental Disorders, Fifth Edition (DSM-5) [1]) diagnosed less than 2 years prior to recruitment. Patients who had significant suicidal risk as assessed by the psychiatrist or diagnosed with dementia, schizophrenia or other psychotic disorder, bipolar I or II disorder, or anxiety disorders including panic disorder, generalized anxiety disorder, obsessive-compulsive disorder and post-traumatic stress disorder, were excluded from the study.

Baseline sociodemographic characteristics as well as comorbidities (such as hypertension, diabetes mellitus, heart disease, stroke, cancer and asthma) were captured. Standardized questionnaires such as Lists of Threatening Events (LTEs) [34] and Temperament and Personality (T&P) [36] were used. The English version of the 12-item LTE questionnaire was reported to have good internal consistency and test-retest reliability (Cronbach's alpha = 0.84; Cohen's kappa = 0.72), with sensitivity of 89% and specificity of 74% [37]; the Malay version by Ng *et al.* (2009) was shown to be reliable as well (Kappa = 0.7 to 0.9) [38].

The T&P questionnaire consisted of 109 questions to capture information on eight personality dimensions and two personality functioning dimensions with Cronbach's α coefficients ranging from 0.62 to 0.91 and intra-class correlations from 0.72 to 0.93 [36, 39]. Items measuring each dimension add up to a scaled score, in which different cut off values were used to determine the tendencies of the dimensions. Cut off points for the different dimensions were as follows: Anxious worrying (18 and above—indicates a greater tendency to become stressed, worried and anxious); Personal reserve (17 and above—is associated with a tendency to keep one's inner feelings to oneself); Perfectionism (31 and above—is associated with a tendency to be very responsible, to have high standards for oneself and to be highly committed to tasks and duties); Irritability (21 and above—is associated with a tendency to be quick-tempered and to externalise stress by becoming snappy and irritated by little things); Social avoidance (17 and above—is associated with a tendency to be introverted); Interpersonal sensitivity (14 and above—is associated with a tendency to worry about rejection or abandonment); Self-criticism (10 and above—is associated with a tendency to be very tough on oneself); Self-focus (9 and over—indicating prioritizing one's own needs over other peoples'); Cooperativeness (20 and above—is associated with a tendency to be generally helpful); and Effectiveness (18 and above—indicates an ability to cope well with different situations and to be confident in problem solving). Higher scores indicate a higher tendency for those dimensions. A clinical record form endorsed by a psychiatrist (NI) (S1 Appendix) was used to identify recurrence based on the definition mentioned above and the symptoms reported at each visit, time from first diagnosis to occurrence of first episode of recurrence of MDD, duration of first episode of recurrence, subsequent episodes of MDD following the first episode of recurrence, severity of first and subsequent MDDs, full or partial remission for each recurrent episode, and family history of MDE. Severity was classified as mild, moderate and severe, according to the number symptoms based on ICD-10 [40], taking into consideration the level of severity of those symptoms and the degree of functional disability using DSM-5 [1]. Mild MDD was defined as presence of two or three MDD symptoms, which were distressing but manageable with minor functional impairment; moderate MDD was characterized by four or more MDD symptoms and patients were functionally impaired; severe MDD was defined by several marked and distressing MDD symptoms which were seriously distressing and unmanageable, with severe functional impairment associated with loss of self-esteem and ideas of worthlessness or guilt, suicidal

thoughts or acts [1]. Full remission is defined by DSM-5 as having no significant signs or symptoms of MDD in the past 2 months [1]. Partial remission is defined as presence of symptoms of the immediate previous MDE but they do not fulfil full criteria for MDD or "there is a period lasting less than 2 months without any significant symptoms of MDE following the end of such an episode" [1].

**Blood collection.** 5 mL of blood was collected using evacuated ethylenediamine tetra acetic acid (EDTA) tubes (Vacutainer Tubes, Becton-Dickinson, NJ, USA) from each subject, was stored under 4˚C and processed on the same day.

**Extraction of genomic DNA.** We used the QIAamp DNA Mini Kit (QIAgen, USA) to extract genomic DNA [41]. 25μL of proteinase K and 200μL of lysis buffer were added to the buffy coat and incubated at 56˚C for 30 minutes to maximize the cell lysis. We then precipitated DNA in 200μL of absolute ethanol before spinning through a filtered spin column. The column was washed twice in AW1 and AW2 wash buffer, followed by a drying spin at maximum speed for 1 minute. DNA was eluted with 50μL of nuclease free water and DNA yield. We then purified and quantified the extracted genomic DNA using a NanoPhotometer®Classic (Implen, USA). DNA purity of all samples were assessed using the 260/280nm ratio within the range of 1.7–2.0.

**Determination of Vitamin D Receptor, zinc transporter-3, Dopamine Transporter-1, Brain-derived neurotropic factor, serotonin receptor 1A and 2A Single Nucleotide Polymorphisms (SNPs).** Primers (both sense and antisense) were used to amplify the targeted single nucleotide polymorphisms (SNPs) namely, *Bsm*I +63980 G>A (*rs1544410)*, *Apa*I +-64978C>A (*rs7975232*) and *Taq*I +65058 T>C (*rs731236*) in the *Vitamin D Receptor* (*VDR) gene, SLC30A3 rs11126396* of ZnT3 gene, *rs40184* C>T of Dopamine Transporter-1 (DAT1) gene, *rs6265* G196A of Brain-derived neurotropic factor (BDNF) gene, *rs6295* -1019C>G of serotonin receptor 1A(HT1A) gene, and *rs6311* -1438A>G of serotonin receptor 2A (HT2A) gene. Details of the genotyping methods have been fully described elsewhere [33, 42] and in S2 Appendix and S3 Appendix.

## Data analysis

Relative frequencies were used to describe variables studied including sociodemographic factors and distribution of single nucleotide polymorphisms of all the genes. Chi-square test or Fisher's exact test were used to determine differences in sociodemographic variables between recurrent patients and non-recurrent patients. Kaplan-Meier analysis was used to perform univariate analysis for sociodemographic variables, family history of psychiatric illnesses, severity of first MDE, comorbidities, and SNPs of *Bsm*I (*rs1544410*), *Taq*I (*rs731236*) and *Apa*I (*rs7975232)* of *VDR* gene, *SLC30A3 rs11126396* of *ZnT3* gene, *rs40184* of *DAT1* gene, *rs6265* of *BDNF* gene, *rs6295* of HT1A, r*s6311* HT2A gene, LTE and T&P domains. Subject(s) who did not experience recurrent MDE at the end of the study period were censored. From the Kaplan-Meier results, variables with p values less than 0.25 were then selected for the Cox regression model which estimated hazard ratios [43, 44, 45].

The Cox-proportionate hazards model was used in regressing time to recurrence on severity of first MDE, social avoidance, irritability, anxious worrying, interpersonal sensitivity, self-criticism, effectiveness, age of first MDE, gender, educational level, family history of psychiatric illness, and comorbidities. Interactions between the significant predictor variables and genotypes of single nucleotide polymorphisms of VDR genes namely, *Bsm*I +63980 G>A, *Apa*I +64978C>A and *Taq*I +65058 T>C, and *SLC30A3 rs11126396* of *ZnT3* gene, *rs40184* of *DAT1* gene, *rs6265* of *BDNF* gene, *rs6295* of HT1A and *rs6311* HT2A gene were tested. Statistical significance was set at alpha of 0.05.

## Results

Over the five-year period, of the 201 patients traced from the case histories, 145 patients were available for survival analysis. 20.9% of the MDD patients suffered first recurrence; the rest were censored. Table 1 shows that there was no difference between the patients with data available for survival analysis and those for whom records were not available. Table 2 depicts the baseline information among those with and without recurrence. The age distribution was similar among both groups. The proportion of patients who experienced recurrence was slightly higher in females compared to males, with a female to male ratio of 1.5:1. Recurrence of MDD was also higher in those with academic degrees and postgraduate qualifications. However, those differences were not statistically significant.

Severity, social avoidance and interaction between *Apa*I polymorphism and severity significantly prognosticated recurrence (Table 3). Individuals with severe MDD at the first MDE had five and a half times higher hazard of developing recurrence compared to mild and moderate MDD (HR = 5.565, 95% CI = 1.631–18.994, p = 0.006). Individuals with a higher score of social avoidance had three and a half times higher hazard of recurrence of MDD (HR = 3.525, 95% CI = 1.349–9.209; p = 0.010). The hazard functions for severity and social avoidance are depicted in Fig 1 and Fig 2 respectively.

*Apa*I +64978C>A was shown to significantly potentiate the effect of severity (Table 4). For A/A genotype, the hazard of recurrence increased by 6.4 times when the severity shifted from mild and moderate to severe; similarly, for the C/A genotype, the hazard ratio increased by a factor of 11.3.

## Discussion

Accurate prediction and identification of factors leading to recurrence of MDD is important to improve customized management to the individual patients. However, accurate ascertainment of the diagnosis is also important before identifying those factors. A number of reviews and research articles used definitions of remission, recovery, relapse and recurrence by Frank *et al.* [46] but the definitions in DSM-5 [1] are clinically more pragmatic and useful, hence these were used in the current study.

From our study, the proportion of first recurrence of MDD was 20.9%. Hardeveld and colleagues reported that the rate of recurrence in specialised mental health care centres was 26.8% [47]. In another article, Hardeveld *et al.* reported that the rate of recurrence over a 5-year period was 60% [16]. According to the American Psychiatric Association [1], there was at least a 60% lifetime risk of recurrence after the first major depressive episode. On the other hand, the 4.3% recurrence rate reported by ten Have *et al.* [48] was much lower than those from this and other studies.

Our study showed that depressed individuals with a higher tendency towards social avoidance were at greater risk of recurrence of MDD. Studies regarding the influence of personality functioning on recurrence of MDD is limited. To the best of our knowledge, no previous study has reported on social avoidance as a predictor of recurrence of MDD. Empirical evidence has been accumulated to show that personality functioning is not only a modifier or a sequela of depressive disorder, but also a predictor of its incidence [49], relapses and recurrences [50, 51]. Ferster (1973) suggested that avoidance plays a role in which frequent avoidance of perceived unpleasant conditions by depressed individuals results in limited exposure to positively-reinforced behaviors and social activities [52, 53] hence increasing the risk of recurrence. In addition, anticipation of previously experienced discrimination may further cause depressed individuals to avoid participation in certain life areas, leading to greater isolation and social marginalization [54]. This finding from our study could potentially be useful

**Table 1. Comparison of cases included for analysis versus missing cases.**

| Variables | Followed up | | Missing | | Statistic[a] | p- value |
|---|---|---|---|---|---|---|
| | **n** | **%** | **n** | **%** | | |
| **Race** | | | | | 1.354 | 0.508 |
| Malay | 100 | 51.0 | 49 | 47.1 | | |
| Chinese | 60 | 30.6 | 30 | 28.8 | | |
| Indian and others | 36 | 18.4 | 25 | 24.0 | | |
| **Gender** | | | | | 0.765 | 0.382 |
| Male | 60 | 30.6 | 37 | 35.6 | | |
| Female | 136 | 69.4 | 67 | 64.4 | | |
| **Education level** | | | | | 0.638 | 0.888 |
| Primary or | 95 | 48.5 | 52 | 50.0 | | |
| Secondary | | | | | | |
| Certificate | 18 | 9.2 | 7 | 6.7 | | |
| Diploma | 30 | 15.3 | 15 | 14.4 | | |
| Degree/Postgraduate | 53 | 27.0 | 30 | 28.8 | | |
| **Comorbidities** | | | | | 0.004 | 0.949 |
| Yes | 69 | 35.2 | 37 | 35.6 | | |
| No | 127 | 64.8 | 67 | 64.4 | | |
| **Family history** | | | | | 0.918 | 0.632 |
| Yes | 54 | 27.6 | 26 | 25.0 | | |
| No | 124 | 63.2 | 71 | 68.3 | | |
| Unsure | 18 | 9.2 | 7 | 6.7 | | |
| **TaqI** | | | | | 0.693 | 0.707 |
| T/T | 139 | 70.9 | 78 | 75.0 | | |
| T/C | 44 | 22.4 | 21 | 20.2 | | |
| C/C | 13 | 6.6 | 5 | 4.8 | | |
| **ApaI** | | | | | 0.123 | 0.940 |
| A/A | 41 | 20.9 | 20 | 19.2 | | |
| A/C | 75 | 38.3 | 41 | 39.4 | | |
| C/C | 80 | 40.8 | 43 | 41.4 | | |
| **Anxious worrying** | | | | | 0.75 | 0.386 |
| High tendency | 51 | 26.6 | 21 | 21.9 | | |
| Low tendency | 141 | 73.4 | 75 | 78.1 | | |
| **Irritability** | | | | | 2.963 | 0.085 |
| High tendency | 25 | 13.0 | 20 | 20.8 | | |
| Low tendency | 167 | 87.0 | 76 | 79.2 | | |
| **Social avoidance** | | | | | 1.169 | 0.280 |
| High tendency | 16 | 8.4 | 12 | 12.4 | | |
| Low tendency | 175 | 91.6 | 85 | 87.6 | | |
| **Interpersonal** | | | | | 0.097 | 0.755 |
| **sensitivity** | 57 | 30.5 | 31 | 32.3 | | |
| High tendency | 130 | 69.5 | 65 | 67.7 | | |
| Low tendency | | | | | | |
| **Self-criticism** | | | | | 0.629 | 0.428 |
| High tendency | 140 | 72.9 | 65 | 68.4 | | |
| Low tendency | 52 | 27.1 | 30 | 31.6 | | |
| **Effectiveness** | | | | | 0.129 | 0.719 |
| High tendency | 60 | 33.3 | 29 | 31.2 | | |

(*Continued*)

**Table 1.** (Continued)

| Variables | Followed up | | Missing | | Statistic[a] | p- value |
|---|---|---|---|---|---|---|
| | **n** | **%** | **n** | **%** | | |
| Low tendency | 120 | 66.7 | 64 | 68.8 | | |
| **Mean Age, years (sd)** | 196 | 38.9 (13.01) | 104 | 38.5 (11.93) | 0.228* | 0.820 |

[a] The statistic reported for all variables was the chi-square value, except the variable labelled * (t-statistic)

for application in clinical practice; by identifying MDD patients with a high tendency to social avoidance, targeted interventions could be undertaken by the psychiatrist or clinical psychologist to mitigate the effect of this modifiable risk factor.

Similar to the results of our study, Kudo *et al.* (2017) did not observe a significant association between the self-criticism personality domain and recurrence of MDD [55]. However, the tools of measurement of the personality dimension (such as anxious worrying, personal reserve and perfectionism) and personality functioning (such as cooperativeness and effectiveness) in both studies were slightly different (Black Dog Institute, 2017). Individuals with a higher tendency towards neuroticism, which was not measured in our study, were being consistently reported to be associated with a higher risk of recurrence [56, 57].

Our study found that severity at first MDE increased hazard of recurrence by five and a half times. However, population and clinical studies that have been conducted used different classification systems for severity of first MDE which involved the duration of the first major depressive episode and number of symptoms present, but a more severe first MDE has consistently been shown to be related to recurrence [22]. Barkow *et al.* (2003) used ICD-10 criteria for severity to study the risk factors of recurrent MDE in the primary care setting and showed that severe MDD at baseline tripled the risk for development of subsequent depressive episodes [58]. NEMESIS-2, a Dutch cohort study using the Sheehan Disability Scale (SDS) to classify severity of depression, showed that severity of the last depressive episode increased recurrence of MDD by almost two times [48]. The results consistently suggested that severity of previous MDE predicted recurrence and this was also demonstrated in our study.

In terms of age at onset and gender, the results of our study replicated results of several earlier studies showing that age at onset of first MDE [17, 18, 59–60] and gender [16, 18, 59–61] did not predict subsequent episode of MDE. This is contrary to what some other studies have

**Table 2. Sociodemographic characteristics of study population by recurrence (n = 145).**

| Variables | Recurrence n (%) | | No Recurrence n (%) | | $\chi^2$ | p value |
|---|---|---|---|---|---|---|
| **Age (years old)** | | | | | 0.017 | 0.898 |
| 18–39 | 24 | 21.2 | 89 | 78.8 | | |
| 40–65 | 17 | 20.5 | 66 | 79.5 | | |
| **Race** | | | | | 1.051 | 0.305 |
| Malay | 18 | 18.0 | 82 | 82.0 | | |
| Others | 23 | 24.0 | 73 | 76.0 | | |
| **Gender** | | | | | 1.831 | 0.176 |
| Male | 9 | 15.0 | 51 | 85.0 | | |
| Female | 32 | 23.5 | 104 | 76.5 | | |
| **Education level** | | | | | 1.672 | 0.196 |
| Primary /Secondary/Certificate | 20 | 17.7 | 93 | 82.3 | | |
| Diploma/Degree/Postgraduate | 21 | 25.3 | 62 | 74.7 | | |

**Table 3. Cox proportionate hazard analysis for predictors of MDD recurrence (n = 145).**

| Variables | | Recurrence n (%) | | No Recurrence n (%) | | Adjusted HR[a] | 95% CI | p value |
|---|---|---|---|---|---|---|---|---|
| Age group | 18–39 | 24 | 21.2 | 89 | 78.8 | 1 | | |
| | 40–65 | 17 | 20.5 | 66 | 79.5 | 0.778 | (0.323–1.874) | 0.575 |
| Gender | Male | 9 | 15.0 | 51 | 85.0 | 1 | | |
| | Female | 32 | 23.5 | 104 | 76.5 | 0.996 | (0.401–2.476) | 0.994 |
| Educational level | Primary / Secondary/Certificate | 20 | 17.7 | 93 | 82.3 | 1 | | |
| | Diploma/Degree/ Postgraduate | 21 | 25.3 | 62 | 74.7 | 1.252 | (0.544–2.883) | 0.597 |
| Comorbidities | Yes | 10 | 14.3 | 60 | 85.7 | 1 | | |
| | No | 31 | 23.7 | 100 | 76.3 | 0.936 | (0.376–2.334) | 0.888 |
| Family History of psychiatric illness | Yes | 45 | 83.3 | 9 | 16.7 | 1 | | |
| | No | 102 | 79.1 | 27 | 20.9 | 1.937 | (0.804–4.666) | 0.141 |
| | Unsure | 13 | 72.2 | 5 | 27.8 | 1.756 | (0.478–6.448) | 0.397 |
| Severity | Mild to Moderate | 20 | 17.4 | 95 | 82.6 | 1 | | |
| | Severe | 21 | 25.9 | 60 | 74.1 | 5.565 | (1.631–18.994) | **0.006*** |
| Anxious Worrying | Low | 27 | 19.0 | 115 | 81.0 | 1 | | |
| | High | 13 | 26.0 | 37 | 74.0 | 0.769 | (0.285–2.071) | 0.603 |
| Irritability | Low | 32 | 19.2 | 135 | 80.8 | 1 | | |
| | High | 7 | 28.0 | 18 | 72.0 | 1.328 | (0.413–4.274) | 0.634 |
| Social Avoidance | Low | 31 | 17.7 | 144 | 82.3 | 1 | | |
| | High | 8 | 50.0 | 8 | 50.0 | 3.525 | (1.349–9.209) | **0.010*** |
| Interpersonal Sensitivity | Low | 27 | 20.6 | 104 | 79.4 | 1 | | |
| | High | 12 | 21.4 | 44 | 78.6 | 0.942 | (0.371–2.394) | 0.900 |
| Self-criticism | Low | 4 | 7.5 | 49 | 92.5 | 1 | | |
| | High | 36 | 25.9 | 103 | 74.1 | 2.964 | (0.810–10.850) | 0.101 |
| Effectiveness | Low | 27 | 22.5 | 93 | 77.5 | 1 | | |
| | High | 10 | 16.7 | 50 | 83.3 | 1.048 | (0.435–2.523) | 0.917 |
| *Taq*I | T/T | 33 | 23.9 | 105 | 76.1 | 1 | | |
| | T/C | 7 | 15.6 | 38 | 84.4 | 0.700 | (0.258–1.900) | 0.483 |
| | C/C | 1 | 7.7 | 12 | 92.3 | 0.141 | (0.016–1.264) | 0.080 |
| *Apa*I*Severity at first MDE | | | | | | | | |
| | *Apa*I*Severity | | | | | 1 | | |
| | [b] *Apa*I (1)*Severity | | | | | 0.582 | (0.181–1.876) | 0.365 |
| | [b] *Apa*I (2)*Severity | | | | | 0.122 | (0.024–0.610) | **0.010*** |

[a]Hazard ratio, controlling for age and gender

[b] Dummy variables: *Apa*I (1) = C/A; *Apa*I (2) = C/C Reference genotype = A/A

*p<0.05

found [7, 11–15]. The inconsistencies in findings regarding age and gender could possibly be due to the effect of socio-cultural differences in the populations studied. In a meta-analysis of the effect of gender and age on depressive symptoms, Salk *et al*. put forward Eagly and Wood's social-structural theory [62] that "*larger* gender differences should be observed in nations with *more* gender inequality". In addition, Salk *et al*. found quadratic trends for gender differences in its interaction with age and age-related factors such as "stressors in adolescence and the hormonal and neurodevelopmental changes that vary by sex and peaking at ages 13–15 years and declining in the 20s". Depression may also manifest differently in different cultures [63]. We postulate that these differences could possibly apply to recurrence, and explain the inconsistent

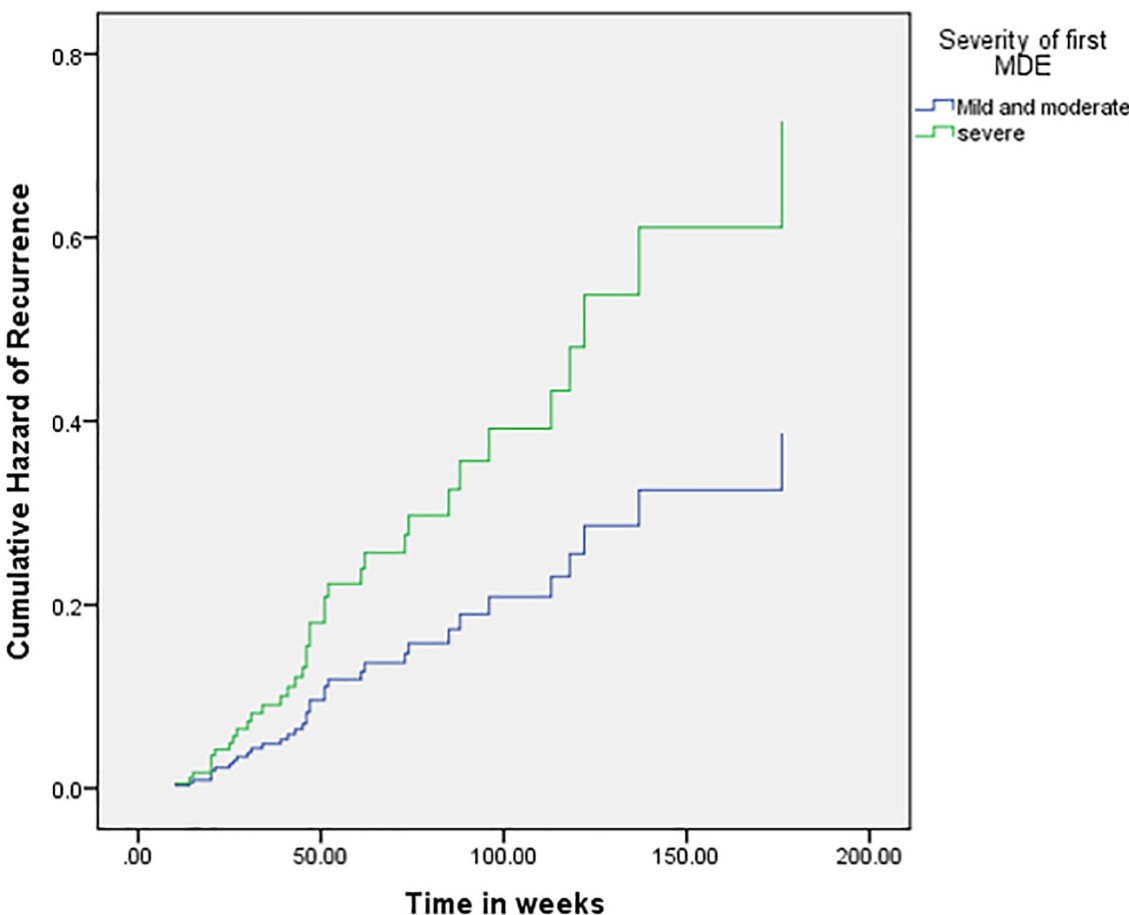

**Fig 1. Hazard function for recurrence of MDD by severity at first MDE.**

findings between studies conducted in different countries and with different age groups studied.

Results regarding family history and recurrence also have been inconsistent–a number of previous studies have reported that a family history of psychiatric illnesses was not associated with recurrence of MDD [11, 44, 64, 65], similar to what we have found in our study. Other studies [21–23] have reported that family history significantly predicted recurrence. In a recent systematic review and meta-analysis, Buckman *et al.* (2018) [66] reported that "there was very limited and inconsistent evidence that family history of depression is prognostic of recurrence". In their meta-analysis, the pooled effect size of an odds ratio of 1.36 (95% CI = 0.92, 2.01) for 3 studies was not statistically significant. The authors reported that "many studies fail to discriminate patients in their first episode from those with a history of multiple previous depressive episodes" and that "differences in the case mix across studies can lead to spurious conclusions". This could have partially accounted for some of the inconsistencies found in the published literature.

Although *Apa*I +64978C>A SNP by itself is not a predictor of recurrence in the present study, we found that the interaction of *Apa*I +64978C>A genotype with severity at first MDE to be significant in predicting its recurrence. The hazard of recurrence for carriers of the A/A and C/A genotypes of *Apa*I with a severe first episode of MDD is much higher than for a less severe episode, indicating that *Apa*I +64978C>A polymorphisms potentiated the effect of

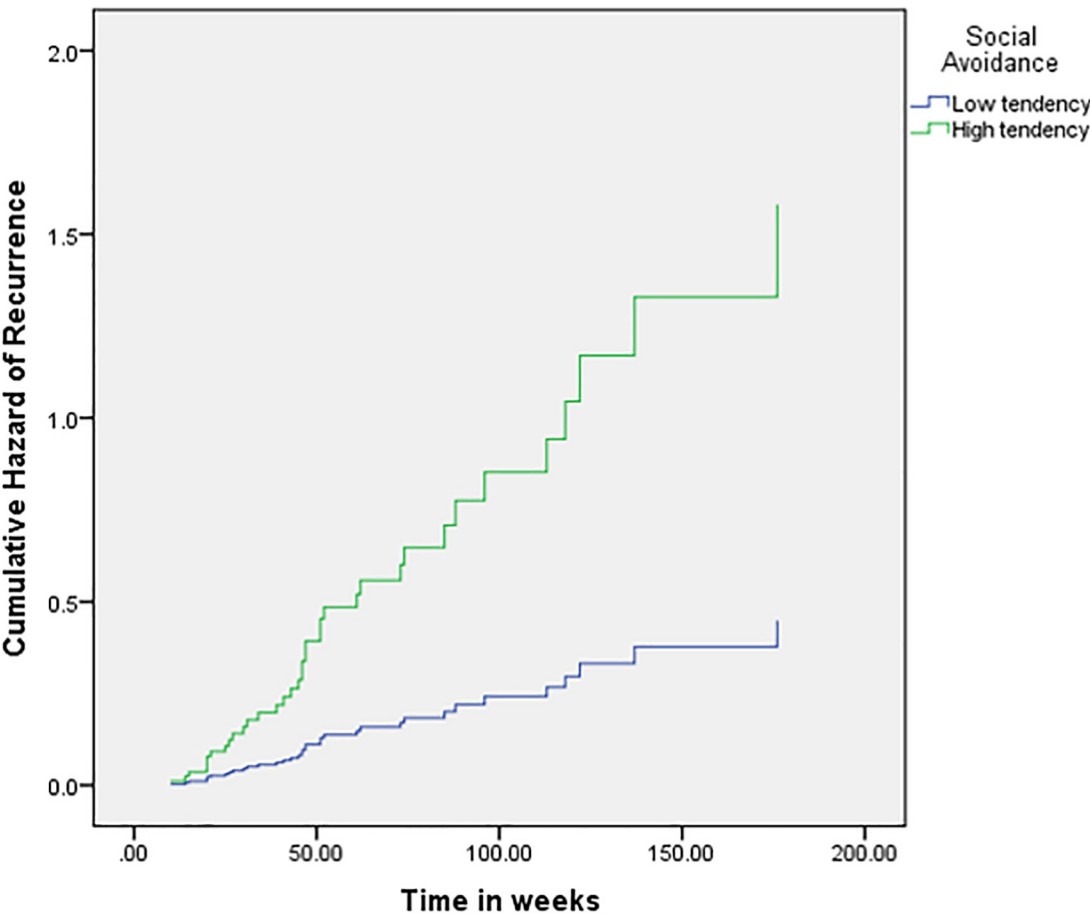

**Fig 2. Hazard function for recurrence of MDD by social avoidance.**

severity on recurrence of MDD (Table 4). Kuningas *et al.* (32) found that Dutch patients with the baT haplotype of the Vitamin D receptor (VDR) gene were less prone to depression compared with carriers of BAt. This propensity of the A allele to increase risk of depression could possibly play a role in the interaction between *Apa*I polymorphism and severity. The C>A transversion at the *Apa*I restriction site is located in the 3' VDR gene transcription initiation site—a ligand binding site of the VDR gene [67]—and may affect the downstream effect of the ligand-binding properties of the Vitamin D receptor, in turn affecting the Vitamin D Response Element (VDRE) complex and activation of the transcription of tryptophan hydroxylase 2 (TPH2) gene, leading to an imbalance of serotonin levels in the brain [68–70]. Although

**Table 4. Hazard ratios for ApaI +64978C>A genotypes by severity at first MDE.**

| | Severity | |
| --- | --- | --- |
| **ApaI** | **Mild and Moderate** | **Severe** |
| A/A | 0.469 (p = 0.239; 95% CI = 0.133–1.656) [*] | 3.018 (p = 0.105, 95% CI = 0.794–14.468) [*] |
| C/A | 0.289 (p = 0.033, 95% CI = 0.092–0.908) [*] | 3.265 (p = 0.041, 95% CI = 1.047–10.182) [*] |
| C/C | 1 | 1 |

[*]Hazard ratio, controlling for age at first MDE

several polymorphisms, including *Apa*I, of the VDR gene have been described, their effects on VDR function and interaction with severity of depression are still poorly understood.

The limitations of the study include those inherent in the design of a case-control study from which data for this study was derived. Biases inherent in a case-control study include recall and selection bias. We attempted to minimize recall bias by administering the questionnaires using enumerators trained in interviewing patients. Since the questionnaires were self-administered, enumerators were on standby to ensure that any questions the patients may have were adequately addressed, with intimations from non-verbal communication from interviewers kept to a minimum. Representativeness of the sample was ensured as far as possible with the recruitment of patients who came from the catchment area of the hospital representative of the cross-section of the community as shown by their socio-demographic profiles (Table 2). The sample size undoubtedly was affected by loss to follow-up; however, there was no significant difference in the baseline profiles between those who were followed up and those lost to follow-up, indicating that the bias could have been kept to a minimum.

This study adds new information to the body of knowledge on modifiable risk factors and predictors for recurrence of MDD; future studies could further examine the role of genetic bio-markers in prognosticating recurrence in assisting health care workers in early identification and preventive management of major depressive disorder. We believe that this new finding has the potential to inform the management of MDD in which clinicians can now be alerted to potential recurrence from a fore knowledge of social avoidance and severity of a previous episode of MDD, both of which are modifiable risk factors.

## Conclusion

This study suggests that social avoidance and severity of previous MDD prognosticate recurrence, with *Apa*I +64978C>A genotypes acting in synergy with severity in potentiating the hazard of recurrence. We suggest screening for personality factors—in particular, social avoidance; this could potentially be performed at the first major depressive episode. Psychological intervention could then be customized to focus on this modifiable factor. In addition, prompt and appropriate management of severe MDD is recommended to reduce risk of recurrence.

## Supporting information

**S1 Appendix.**
(DOCX)

**S2 Appendix. Genotyping of VDR Single Nucleotide Polymorphisms (SNPs).**
(DOCX)

**S3 Appendix. Genotyping for single nucleotide polymorphisms of *HTR1A*-rs6295 and *HTR2A*-rs6311.**
(DOCX)

## Acknowledgments

We thank the Director General of Health Malaysia for approval to conduct the study (NMRR No.: NMRR-14-688-19696). We also thank Dr Azizul Awaluddin, Dr Sharifah Suziah Syed Mokhtar, Dr Mazni Mat Junus, Dr Elinda Tunan, Ms Siti Zubaidah Redzuan, Dr Vaidehi Ulanganathan, research assistants as well as staff of the hospitals involved.

## Author Contributions

**Conceptualization:** Munn-Sann Lye, Normala Ibrahim, King-Hwa Ling, Johnson Stanslas, Su-Peng Loh, Rozita Rosli.

**Data curation:** Yin-Yee Tey, Aisya Farhana Shahabudin, Khairul Aiman Lokman, Ibrahim Mohammed Badamasi, Asraa Faris-Aldoghachi, Nurul Asyikin Abdul Razak.

**Formal analysis:** Munn-Sann Lye, Yin-Yee Tey.

**Funding acquisition:** Munn-Sann Lye, Normala Ibrahim, King-Hwa Ling, Johnson Stanslas, Su-Peng Loh, Rozita Rosli.

**Investigation:** Yin-Sim Tor, Aisya Farhana Shahabudin, Ibrahim Mohammed Badamasi, Asraa Faris-Aldoghachi, Nurul Asyikin Abdul Razak.

**Methodology:** Munn-Sann Lye, Normala Ibrahim, King-Hwa Ling, Johnson Stanslas, Su-Peng Loh, Rozita Rosli.

**Project administration:** Munn-Sann Lye, Normala Ibrahim, King-Hwa Ling, Johnson Stanslas, Su-Peng Loh, Rozita Rosli.

**Resources:** Munn-Sann Lye, Normala Ibrahim, King-Hwa Ling, Johnson Stanslas, Su-Peng Loh, Rozita Rosli.

**Supervision:** Munn-Sann Lye.

**Validation:** Munn-Sann Lye.

**Writing – original draft:** Munn-Sann Lye, Yin-Yee Tey.

**Writing – review & editing:** Munn-Sann Lye, Yin-Yee Tey, Yin-Sim Tor.

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
