## [Decision Letter · Decision Letter 0]

2 Jan 2020

PONE-D-19-27517

Predictors of a first recurrence of major depressive disorder

PLOS ONE

Dear Professor Lye,

Thank you for submitting your manuscript to PLOS ONE. After careful consideration, we feel that it has merit but does not fully meet PLOS ONE’s publication criteria as it currently stands. Therefore, we invite you to submit a revised version of the manuscript that addresses the points raised during the review process.

We would appreciate receiving your revised manuscript by Feb 16 2020 11:59PM. To enhance the reproducibility of your results, we recommend that if applicable you deposit your laboratory protocols in protocols.io, where a protocol can be assigned its own identifier (DOI) such that it can be cited independently in the future. For instructions see: http://journals.plos.org/plosone/s/submission-guidelines#loc-laboratory-protocols

We look forward to receiving your revised manuscript.

Kind regards,

Zezhi Li, Ph.D., M.D.

Academic Editor

PLOS ONE

Journal Requirements:

1.

Reviewers' comments:

Reviewer's Responses to Questions

**Comments to the Author**

1. Is the manuscript technically sound, and do the data support the conclusions?

Reviewer #1: Yes

Reviewer #2: Yes

2. Has the statistical analysis been performed appropriately and rigorously? 

Reviewer #1: No

Reviewer #2: Yes

3. Have the authors made all data underlying the findings in their manuscript fully available?

Reviewer #1: Yes

Reviewer #2: Yes

4. Is the manuscript presented in an intelligible fashion and written in standard English?

Reviewer #1: No

Reviewer #2: Yes

5. Review Comments to the Author

Reviewer #1: The authors want to identify the predictive factors for recurrence of major depressive disorders in 5-year follow-up study. They found that the social avoidance and severity of previous MDD could predict the recurrence, with 287 ApaI +64978C>A genotypes acting in synergy with severity in potentiating the risk of recurrence. It is an interesting paper, but some concerns should be addressed.

In the abstract

1. The aim of the present study should be presented.

2. “201 cases of major depressive disorder from four public hospitals in Malaysia were traced over a 5-year period from 2014 to 2017”. It should be “A total of 201 patients with major depressive disorder from four hospitals in Malaysia were followed up for 5 year”. Case should be revised as “patients” throughout the paper.

In the introduction

3. The definition of “recurrence” should cite the reference.

In the methods

4. Please clarify the inclusion and exclusion criteria for recruiting the patients.

5. Please detail the methods of DNA extraction.

6. Why the authors selected the SNPs of Vitamin D Receptor, Zinc Transporter-3, Dopamine Transporter-1, Brain-derived Neurotropic Factor, Serotonin Receptor 1A and 2A?

7. Why the author selected the variables with p values less than 0.25 from the Kaplan-Meier results, for the next Cox regression model?

In the discussion

8. The author should discuss the limitation of this study, such as the moderate sample size.

9. Overall, the language of the study should be improved, and the paper should be revised carefully.

Reviewer #2: Thank you for the opportunity to submit this manuscript. The information that Social avoidance and vitamin D receptor were prognostic of recurrence of major depressive disorder was very helpful. However, I still have several comments. I also think that some more discussion is warranted. In its current shape I am not fully convinced if this manuscript is ready to be published.

1.I don’t think that paragraph 1 with so much words is essential, considering the major aims of this manuscript.

2.I would suggest that more should be added to paragraph 3 (line 73 to 76 ) in introduction section considering the major findings of this manuscript.

3.Just as the author stated in Introduction section that those of younger age, women and family history of MDD were at higher risk of recurrence. However these findings were not consist with results of this manuscript. Readers would have great interest on the discussion of this part.

4.I would suggest that limitations should be listed in discussion section.

5.It is interesting that the interaction of ApaI +64978C>A genotype with severity at first MDE would predicted the recurrence of MDE. It would be better if more discussion about potential mechanisms added to this part(line 269 to 276) , instead of listing the result.

6.It would be better if more references from the past 5 years are added.

6. PLOS authors have the option to publish the peer review history of their article (what does this mean?). If published, this will include your full peer review and any attached files.

Reviewer #1: No

Reviewer #2: No

---

## [Author Response · Author response to Decision Letter 0]

15 Feb 2020

Response to Reviewers

Reviewer 1:

Abstract

• The aim of the present study should be presented.

The aim of the present study has been included. Pg 2, line 24 - 25

• “201 cases of major depressive disorder from four public hospitals in Malaysia were traced over a 5-year period from 2014 to 2017”. It should be “A total of 201 patients with major depressive disorder from four hospitals in Malaysia were followed up for 5 year”. Case should be revised as “patients” throughout the paper. 

The sentence has been re-written as suggested. Pg 2, line 23 - 24

‘Case’ has been replaced with ‘patients’ where relevant.

Introduction

• The definition of “recurrence” should cite the reference. 

The reference has been cited. [Reference number (1)]

It is “American Psychiatric Association. (2014). Diagnostic and statistical manual of mental disorders. 5th ed. Washington, DC: American Psychiatric Publishing”. Pg 3, line 52 

Methods

• Please clarify the inclusion and exclusion criteria for recruiting the patients.

The inclusion and exclusion criteria have been added. Pg 4 – 5, lines 93 – 99

• Please detail the methods of DNA extraction.

The method of DNA extraction has been added. Pg 7, lines 148 – 156

• Why the authors selected the SNPs of Vitamin D Receptor, Zinc Transporter-3, Dopamine Transporter-1, Brain-derived Neurotropic Factor, Serotonin Receptor 1A and 2A?

Mutations in the brain-derived neurotropic factor (BDNF) [29, 30] and serotonin receptor genes HT1A and HT2A genes [31] were shown to be associated with recurrent MDD. Serotonin receptor 1A and 2A were studied in view of their close relationship with serotonin as a modifier of mood, and mood disturbances occur when there is an imbalance of serotonin metabolism caused by mutations in HT1A and HT2A genes. 

In addition, findings by Lye et al. that single nucleotide polymorphisms (SNPs) of vitamin D receptor (VDR) and SLC30A3 of zinc transporter-3 (ZnT3) genes were associated with MDD [32, S1 Text], and together with evidence that SNPs of dopamine transporter-1 [33, 34] increased risk of MDD, led us to suspect if an association exists between VDR, ZnT3 SNPs and recurrence of MDD as well. Pg 4, lines 73 – 78

• Why the author selected the variables with p values less than 0.25 from the Kaplan-Meier results, for the next Cox regression model? 

Statistically, Kaplan-Meier is a univariate type of analysis – it regresses the outcome (time to recurrence) on one single independent variable - hence the estimates of hazard ratios and p-values derived from each of these Kaplan-Meier runs are therefore not adjusted for confounding. So apart from selecting the obviously significant variables that had p values less than 0.05, the process of selecting those other variables with p-values greater than 0.05 but less than 0.25 is to attempt to ensure that enough leeway is given so that when we run the Cox regression model adjusting for confounders, p-values of some of those selected variables that were not significant initially with univariate analysis, (that is p-values greater than 0.05 but less than 0.25), may now fall below 0.05 after adjusting for the confounding variables in the multivariate model. A number of research articles [appended below] have used 0.25 as the cut-off point for selection of variables to enter into the Cox regression model, while Robinson (2001) provided the statistical rationale involving the cut-off point of the p-value up to a maximum of 0.25. Pg 8, lines 178 – 179

References: 

1. Singh R, Tripathi V, Kalpana Singh MK, Dwivedi SN. Determinants of Birth Intervals in Tamil Nadu in India: Developing Cox Hazard Models with Validations and Predictions. Revista Colombiana de Estadística 2012; 35: 289–307.[42]

2. Sendek EM, Hebo SH. Modeling time-to-good control of hypertension using Cox proportional hazard and frailty models at Bahir-Dar Felege Hiwot Referral Hospital. Open Access Medical Statistics. 2017; 7:27–36. [43]

3. Robinson DH, Wainer H. On the Past and Future of Null Hypothesis Significance Testing. (Research Report No. RR-01-24) Statistics & Research Division Princeton, NJ, USA. Educational Testing Service. 2001; 25. [44]

Result

• The author should discuss the limitation of this study, such as the moderate sample size.

The limitations have been added. Pg 18, lines 319 - 330. 

• Overall, the language of the study should be improved, and the paper should be revised carefully. 

Overall, we have improved on the language, and the paper has been carefully revised according to the reviewer’s comments. 

Reviewer 2:

I don’t think that paragraph 1 with so much words is essential, considering the major aims of this manuscript. 

Paragraph 1 has been shortened, while paragraph 2 has been incorporated in paragraph 1. Pg 3, Lines 48 – 59 

I would suggest that more should be added to paragraph 3 (line 73 to 76 ) in introduction section considering the major findings of this manuscript. 

Paragraph 3, now the new paragraph 2, has been expanded, relevant to the major findings of this study. Pg 3, lines 61 – 71 

Just as the author stated in Introduction section that those of younger age, women and family history of MDD were at higher risk of recurrence. However these findings were not consist with results of this manuscript. Readers would have great interest on the discussion of this part. 

The findings that were inconsistent have been discussed. Pg 16 – 17, lines 277 – 301 

I would suggest that limitations should be listed in discussion section.

The limitations have been included in the discussion section. Pg 18, lines 319 - 330

It is interesting that the interaction of ApaI +64978C>A genotype with severity at first MDE would predicted the recurrence of MDE. It would be better if more discussion about potential mechanisms added to this part (line 269 to 276) , instead of listing the result.

More discussion on the potential mechanisms has been added to this part. Pg 17, lines 303 – 317 

It would be better if more references from the past 5 years are added.

More recent references have been added. 

1. Monroe SM, Harkness KL. Recurrence in major depression: A conceptual analysis. Psychol Rev. 2011; 118: 655–674.

2. Kessler RC, Bromet EJ. The epidemiology of depression across cultures. Annu Rev Public Health. 2013; 34: 119–38. 

3. Lee AS. Better outcomes for depressive disorders? Psychol Med. 2003; 33: 769–774. 

4. Gueorguieva R, Chekroud AM, Krystal JH. Trajectories of relapse in randomised, placebo-controlled trials of treatment discontinuation in major depressive disorder: an individual patient-level data meta-analysis. Lancet Psy. 2017; 4:230–237. 

5. Serafini G, Santi F, Gonda X, Aguglia A, Fiorillo A, Pompili M, F.Carvalhoh A, Amore M. Predictors of recurrence in a sample of 508 outpatients with major depressive disorder. J Psy Research. 2019; 114: 80-87.

6. Hardeveld F, Spijker J, De Graaf R, Nolen WA, Beekman AT. Prevalence and predictors of recurrence of major depressive disorder in the adult population. Acta Psychiatr Scand. 2010; 122(3): 184 – 191.

7. Lewinsohn PM, Zeiss AM, Duncan EM. Probability of relapse after recovery from an episode of depression. J Abnormal Psychol. 1989; 98(2): 107-116.

8. Kovac M, Obrosky DS, Sherrill J. Developmental changes in the phenomenology of depression in girls compared to boys from childhood onward. J Affect Disord. 2003;74: 33−48.

9. Lewinsohn PM, Rohde P, Seeley JR, Klein DN, Gotlib IH. Natural course of adolescent major depressive disorder in a community sample: Predictors of recurrence in young adults. Am J Psychiatry. 2000; 157:1584 – 1591.

10. Serafini G, Nebbia J, Cipriani N, Conigliaro C, Erbuto D, Pompili M, Amore M. Number of illness episodes as predictor of residual symptoms in major depressive disorder. Psychiatr Res. 2018; 262: 469–476.

11. Nierenberg AA, Husain MM, Trivedi MH, Fava M, Warden D, Wisniewski SR, Miyahara S, Rush AJ. Residual symptoms after remission of major depressive disorder with citalopram and risk of relapse: a STAR*D report. Psychol Med. 2010; 40: 41–50.

12. Nery FG, Hatch JP, Nicoletti MA, Monkul ES, Najt P, Matsuo K, Cloninger CR, Soares JC. Temperament and character traits in major depressive disorder: influence of mood state and recurrence of 

episodes. Depress Anxiety. 2009;26(4):382-8

13. Nöbbelin L, Bogren M, Mattisson C, Brådvik L. Risk factors for recurrence in depression in the Lundby population, 1947–1997. J Affect Disord. 2008; 228: 125–131.

14. Wood W, Eagly AH. Biosocial construction of sex differences and similarities in behavior. Advances in Experimental Social Psychology. 2012; 46: 55–123.

15. Tsai JL, Chentsova-Dutton Y. Understanding depression across cultures. In Gotlib IH, Hammen CL, editors. Handbook of depression. New York: Guilford Press; 2002. pp. 467–491.

16. Buckman JEJ, Underwood A, Clarke K, Saunders R, Hollon SD, Fearon P, Pilling S. Risk factors for relapse and recurrence of depression in adults and how they operate: A four-phase systematic review and meta-synthesis. Clin Psychol Rev. 2018; 64:13-38.

---

## [Decision Letter · Decision Letter 1]

28 Feb 2020

Predictors of recurrence of major depressive disorder

PONE-D-19-27517R1

Dear Dr. Lye,

We are pleased to inform you that your manuscript has been judged scientifically suitable for publication and will be formally accepted for publication once it complies with all outstanding technical requirements.

With kind regards,

Zezhi Li, Ph.D., M.D.

Academic Editor

PLOS ONE

Reviewers' comments:

Reviewer's Responses to Questions

**Comments to the Author**

1. If the authors have adequately addressed your comments raised in a previous round of review and you feel that this manuscript is now acceptable for publication, you may indicate that here to bypass the “Comments to the Author” section, enter your conflict of interest statement in the “Confidential to Editor” section, and submit your "Accept" recommendation.

Reviewer #1: All comments have been addressed

Reviewer #2: All comments have been addressed

2. Is the manuscript technically sound, and do the data support the conclusions?

Reviewer #1: Yes

Reviewer #2: Yes

3. Has the statistical analysis been performed appropriately and rigorously? 

Reviewer #1: Yes

Reviewer #2: Yes

4. Have the authors made all data underlying the findings in their manuscript fully available?

Reviewer #1: Yes

Reviewer #2: Yes

5. Is the manuscript presented in an intelligible fashion and written in standard English?

Reviewer #1: Yes

Reviewer #2: Yes

6. Review Comments to the Author

Reviewer #1: (No Response)

Reviewer #2: It has clearly improved, although better it can be.I would suggest that the edior could consider to publish this manuscript .

7. PLOS authors have the option to publish the peer review history of their article (what does this mean?). If published, this will include your full peer review and any attached files.

Reviewer #1: No

Reviewer #2: No

---

## [Editor Report · Acceptance letter]

4 Mar 2020

PONE-D-19-27517R1 

Predictors of recurrence of major depressive disorder 

Dear Dr. Lye:

I am pleased to inform you that your manuscript has been deemed suitable for publication in PLOS ONE. Congratulations! Your manuscript is now with our production department. 

With kind regards,

on behalf of

Dr. Zezhi Li 

Academic Editor

PLOS ONE